# Tofogliflozin Delays Portal Hypertension and Hepatic Fibrosis by Inhibiting Sinusoidal Capillarization in Cirrhotic Rats

**DOI:** 10.3390/cells13060538

**Published:** 2024-03-19

**Authors:** Shohei Asada, Kosuke Kaji, Norihisa Nishimura, Aritoshi Koizumi, Takuya Matsuda, Misako Tanaka, Nobuyuki Yorioka, Shinya Sato, Koh Kitagawa, Tadashi Namisaki, Takemi Akahane, Hitoshi Yoshiji

**Affiliations:** Department of Gastroenterology, Nara Medical University, Kashihara 634-8521, Nara, Japan; asahei@naramed-u.ac.jp (S.A.); yoshijih@naramed-u.ac.jp (H.Y.)

**Keywords:** liver sinusoidal endothelial cell, hepatic stellate cell, sodium glucose transporter 2 inhibitors, oxidative stress, angiogenesis

## Abstract

Background: Liver cirrhosis leads to portal hypertension (PH) with capillarization of liver sinusoidal endothelial cells (LSECs), although drug treatment options for PH are currently limited. Sodium glucose transporter 2 inhibitors, which are antidiabetic agents, have been shown to improve endothelial dysfunction. We aimed to elucidate the effect of tofogliflozin on PH and liver fibrosis in a rat cirrhosis model. Methods: Male-F344/NSlc rats repeatedly received carbon tetrachloride (CCl_4_) intraperitoneally to induce PH and liver cirrhosis alongside tofogliflozin (10 or 20 mg/kg). Portal hemodynamics and hepatic phenotypes were assessed after 14 weeks. An in vitro study investigated the effects of tofogliflozin on the crosstalk between LSEC and activated hepatic stellate cells (Ac-HSC), which are relevant to PH development. Results: Tofogliflozin prevented PH with attenuated intrahepatic vasoconstriction, sinusoidal capillarization, and remodeling independent of glycemic status in CCl_4_-treated rats. Hepatic macrophage infiltration, proinflammatory response, and fibrogenesis were suppressed by treatment with tofogliflozin. In vitro assays showed that tofogliflozin suppressed Ac-HSC-stimulated capillarization and vasoconstriction in LSECs by enhancing the antioxidant capacity, as well as inhibited the capilliarized LSEC-stimulated contractive, profibrogenic, and proliferative activities of Ac-HSCs. Conclusions: Our study provides strong support for tofogliflozin in the prevention of liver cirrhosis-related PH.

## 1. Introduction

Advanced liver fibrosis and portal hypertension (PH) are serious consequences of chronic liver disease, the complications of which (e.g., esophagogastric variceal bleeding, hepatic encephalopathy, and ascites retention) can lead to a decompensated state [1,2]. Two mechanisms are mainly behind PH: increased intrahepatic vascular resistance (IHVR) and splanchnic vasodilatation with increased portal blood inflow [3,4]. In the fibrotic liver, IHVR is attributed to both structural remodeling of the liver architecture and increased intrahepatic vascular tone, which are caused by the activation of hepatic stellate cells (HSCs) and dysfunction of liver sinusoidal endothelial cells (LSECs) [3,4].

Prior to liver fibrosis, LSECs undergo a dysfunctional process characterized by gradual loss of their characteristic transmembrane pores (i.e., fenestrae) and increased basement lamina deposition (i.e., sinusoidal capillarization), which contributes to increased IHVR [3,4,5]. In the cirrhotic liver, intrahepatic vasoconstriction is promoted by an increase in vasoconstrictors, such as endothelin-1 (ET-1), and a decrease in vasodilators, such as nitric oxide (NO) produced by an endothelial nitric oxide synthase (eNOS) [3,4,5]. In addition, intrahepatic pathological angiogenesis, which is mediated by platelet-derived growth factor (PDGF), vascular endothelial growth factor (VEGF), and angiopoietin (Ang), plays a central role in increased splanchnic blood flow and the formation of collaterals in liver cirrhosis [6].

Activated HSC (Ac-HSC) is a representative cell component involved in liver fibrogenesis; it disturbs the balance of the extracellular matrix and promotes the secretion of proinflammatory cytokines [3,4,5]. Advanced fibrosis compresses the hepatic sinusoids and distorts the normal liver structure, leading to increased IHVR [3,4,5]. Another element of IHVR is HSC contractility, mainly regulated by ET-1 signaling, which is significant since HSCs reside in the space of Disse, surrounding the hepatic sinusoids [3,4,5,6,7,8,9]. Notably, the crosstalk between LSECs and HSCs plays a crucial role in the pathogenesis of PH. In particular, loss of the LSEC phenotype promotes HSC activation [3,4,5,6,7,8,9,10].

Currently, nonselective beta-blockers are the pharmaceutical approach recommended by the Baveno VII consensus, but it is not universally effective in cirrhotic patients with clinically significant PH [11]. Thus, there is an urgent need to identify a clinically available compound that improves PH in liver cirrhosis with proven safety for long-term administration.

Sodium glucose transporter 2 inhibitors (SGLT2-Is) are antidiabetic agents that act on SGLT2 in the proximal tubule to facilitate natriuresis and glucosuria [12]. They are also emerging as promising novel agents against metabolic dysfunction-associated steatotic liver disease (MASLD) [13,14,15]. Recently, SGLT2-Is were demonstrated to have multiple ancillary effects; these are now prescribed even in nondiabetic patients for their clinical benefits in heart failure and chronic kidney disease [16,17]. Recent case reports have demonstrated that SGLT2-Is reduce ascites and peripheral edema in patients with liver cirrhosis, which can be attributed to their action on endothelial cells [18]. Several studies have shown that SGLT2-Is can directly affect endothelial function independently of glycemic status and through various pathways, resulting in improved vasodilation via increased NO production and bioavailability [19,20,21]. Moreover, SGLT2-Is also repressed ET-1 expression while enhancing eNOS activation in a mouse myocardial ischemia–reperfusion injury model [22].

Based on this preliminary evidence, we hypothesized that SGLT2-Is would have the potential to reduce PH and liver fibrosis progression by rearrangement of the crosstalk between LSECs and HSCs in liver cirrhosis. This study aimed to elucidate the effects of tofogliflozin, one of the SGLT2-Is, on PH and liver fibrosis in a carbon tetrachloride (CCl_4_)-induced cirrhotic rat model.

## 2. Materials and Methods

### 2.1. Animals and Compounds

Ten-week-old male F344/NSlc rats (Japan SLC, Hamamatsu, Japan) were caged with free access to food and water and with controlled temperature (23 ± 3 °C) and humidity (50 ± 20%) and a 12 h light/dark cycle. This study was reviewed and approved by the ethics committee of Nara Medical University (No. 13136) and was performed in accordance with the Guide for Care and Use of Laboratory Animals of the National Research Council. Tofogliflozin was generously provided by Kowa Co., Ltd. (Tokyo, Japan).

### 2.2. In Vivo Experimental Protocol

Rats (*n* = 40) were randomly divided into four experimental groups (*n* = 10) and underwent treatment for 14 weeks as follows (Figure 1A): (i) intraperitoneal injection of corn oil twice a week and oral gavage of saline every day as a vehicle (C/O group), (ii–iv) intraperitoneal injections of CCl_4_ (0.5 mL/kg) diluted 15% with corn oil (Nacalai Tesque, Kyoto, Japan) twice a week and oral gavage of (ii) saline every day (Veh group), (iii) low-dose tofogliflozin (10 mg/kg) every day (10 mg group), and (iv) high dose of tofogliflozin (20 mg group) [23]. After the 14-week experimental period, body and liver weights, portal dynamics, and systemic dynamics were measured; then, blood was collected from the aorta to measure serum levels of hepatic enzymes, kidney function test, and glucose. Urine was collected from the bladder to measure urine glucose levels.

### 2.3. In Vivo Hemodynamic Evaluation

Experimental rats were given inhalation anesthesia with isoflurane (4–5% for induction and 1–2% for maintenance) (Merck & Co., Rahway, NJ, USA). The femoral artery and ileocolic vein were cannulated with a Transonic Science Pressure Catheter (FTH-1211B-0018; Transonic Science, London, ON, Canada) to measure portal vein pressure (PVP) (mmHg), mean arterial pressure (MAP) (mmHg), and heart rate (HR) (bpm), which were analyzed using a Transonic Science Pressure System (FP095B; Transonic Science). Blood flow measurement probes (MA2PSB; Transonic Science) were placed around the portal vein, as close as possible to the liver, to measure portal blood flow (PBF) (mL/min). Intrahepatic vascular resistance (IHVR, mmHg/mL/min^−1^) was calculated as PVP/PBF. The temperature of the animals was maintained at 37 ± 0.5 °C. Hemodynamic data were collected after a 20 min stabilization period.

### 2.4. Cell Culture

Human LSECs (hLSECs; TMNK-1), human hepatocytes (hHCs; HepaMN), and human umbilical vein endothelial cells (hUVECs; HUEhT-1) were purchased from the National Institutes of Biomedical Innovation, Health and Nutrition (Osaka, Japan). The activated human HSC (Ac-hHSC) line, LX-2, was purchased from Merck KGaA (Darmstadt, Germany). TMNK-1 was suspended in Complete Medium Kit with Serum and Culture Boost-R (Cell Systems, Kirkland, WA, USA), while LX-2, HepaMN, and HUEhT-1 were suspended in Dulbecco’s modified Eagle’s medium (DMEM) (Nacalai tesque). These were supplemented with 2% fetal bovine serum (FBS) (Gibco, Waltham, MA, USA) and antibiotics (1% penicillin and streptomycin), plated on 100 mm plastic culture dishes, and incubated at 37 °C in a 5% CO_2_ air environment. All cell lines were authenticated using Short Tandem Repeat profiling within the last 3 years. *Mycoplasma* testing was performed using the MycoProbe^®^ Mycoplasma Detection Kit (R&D Systems, Minneapolis, MN, USA) according to the manufacturer’s protocol.

### 2.5. Statistical Analyzes

Data are presented as their mean ± SD. All experiments were performed a minimum of two times unless otherwise stated. Two-tailed Student’s *t*-test and Mann–Whitney *U* test for some assays were used to analyze statistical differences using GraphPad Prism version 9.0 (GraphPad Software, La Jolla, CA, USA), with statistical significance set at * *p* < 0.05 and ** *p* < 0.01.

Additional methods can be found online in the Appendix A.

## 3. Results

### 3.1. Tofogliflozin Prevents Portal Hypertension Development in CCl_4_-Treated Rats

Figure 1A shows the effects of tofogliflozin on liver cirrhosis induced by 14 weeks of CCl_4_ treatment in rats. Body weight was remarkably decreased after chronic CCl_4_ treatment. Tofogliflozin tended to ameliorate this decrease in body weight but with no significant difference in both doses (10 and 20 mg/kg) (Figure 1B). Likewise, the relative liver weight was not significantly changed after tofogliflozin treatment (Figure 1C). As shown in Figure 1D, urinary glucose levels were increased in tofogliflozin-treated rats, but serum glucose levels were not altered by tofogliflozin treatment (Figure 1E). We found no evidence of renal impairment with CCl_4_ and/or tofogliflozin administration (Figure 1F).

Next, we assessed the effect of tofogliflozin on PH. CCl_4_-treated rats had significantly higher PVP, decreased PBF, and higher IHVR than corn oil-treated control rats (Figure 1G–I). From a PVP of 13.2 ± 1.5 mmHg, treatment with 10 and 20 mg/kg of tofogliflozin reduced this to 9.9 ± 1.5 and 7.9 ± 1.0 mmHg, respectively (Figure 1G). Tofogliflozin treatment also increased PBF and consequently decreased IHVR in CCl_4_-treated rats (Figure 1H,I). Notably, in the tofogliflozin-treated group, the decrease in PVP and IVHR was more pronounced in the 20 mg/kg-treated group (Figure 1G,I). CCl_4_-treated rats showed a significantly lower MAP compared with corn oil-treated control rats, whereas MAP and HR remained unchanged after tofogliflozin treatment (Figure 1J,K).

### 3.2. Tofogliflozin Attenuates Intrahepatic Vasoconstriction and Vascular Resistance in CCl_4_-Treated Rats

In cirrhosis, hepatic vascular dysfunction contributes to PH; the hepatic vasculature is unable to properly adapt its tone, favoring vasoconstriction over vasodilation. This was seen in CCl_4_-treated rats, as evidenced by the elevated hepatic levels of ET-1 as well as decreased levels of eNOS, NO, and cyclic guanosine monophosphate (cGMP) (Figure 2A–D). Notably, treatment with tofogliflozin significantly reduced ET-1 and increased eNOS, NO, and cGMP in the livers of CCl_4_-treated rats (Figure 2A–D). In the tofogliflozin-treated group, the decrease in ET-1 and the increase in eNOS were more pronounced in the 20 mg/kg-treated group (Figure 2A,B). We further evaluated the effect of tofogliflozin on the hepatic expression of vascular tone-related markers, including cystathionine γ-lyase (CTH) and dimethylarginine dimethyl amino-hydrolase 1 (DDAH1). CTH is an enzyme that produces hydrogen sulfide, which is an endogenous gaseous vasodilator. Conversely, DDAH1 is a metabolizing enzyme of asymmetric dimethyl arginine, an endogenous inhibitor of eNOS [24]. Hepatic mRNA levels of *Cth* and *Ddah1* were increased after tofogliflozin treatment at both doses (10 and 20 mg/kg) (Figure 2E). GTP cyclohydrolase (GCH1) was also increased; this is the rate-limiting enzyme for the production of tetrahydrobiopterin, a crucial cofactor of eNOS (Figure 2F).

### 3.3. Tofogliflozin Inhibits Sinusoidal Capillarization and Remodeling in CCl_4_-Treated Rats

Sinusoidal capillarization plays a key role in tissue remodeling and fibrosis in liver cirrhosis. Therefore, we investigated the change in angiogenic status after treatment with tofogliflozin in CCl_4_-treated rats. As shown in Figure 2G,H, newly formed CD34^+^ intrahepatic vessels were increased alongside liver fibrosis in CCl_4_-treated rats, indicating the development of sinusoidal capillarization. Tofogliflozin treatment remarkably attenuated CD34^+^ sinusoidal capillarization to approximately 30% of that in the vehicle-treated group (Figure 2G,H). In line with this, the hepatic levels of VEGF-A and PDGF-BB were significantly reduced (Figure 2I,J). Furthermore, hepatic von Willebrand factor (vWF), an endothelial activation marker, was significantly decreased after tofogliflozin treatment (Figure 2K). The suppression of sinusoidal capillarization after tofogliflozin treatment was also confirmed by a marked decrease in the mRNA expression of *Vegfa*, *Vwf*, *Ang1*, and *Ang2* (Figure 2L).

### 3.4. Tofogliflozin Suppresses Hepatic Necroinflammation and Macrophage Infiltration in CCl_4_-Treated Rats

Notably, treatment with tofogliflozin significantly inhibited the elevation of transaminases, especially at 20 mg/kg, in CCl_4_-treated rats (Figure 3A). Accordingly, hematoxylin and eosin staining showed a reduction in necroinflammation after treatment with tofogliflozin (Figure 3B). Regarding the markers of macrophage infiltration, there was a marked increase in hepatic CD68-positive Kupffer cell expansion and *Adgre1* mRNA expression, which were significantly attenuated by tofogliflozin treatment at both doses (10 and 20 mg/kg) (Figure 3C–E). In accordance with the reduction in Kupffer cell infiltration, tofogliflozin treatment decreased the hepatic mRNA expression of proinflammatory cytokines, including *Tnfa*, *Il6*, and *Il1b* (Figure 3F).

### 3.5. Tofogliflozin Exerts Antifibrotic Effects in CCl_4_-Treated Rats

CCl_4_-treated rats exhibited cirrhotic livers, as identified by Sirius-Red staining of liver sections (Figure 4A,B). Treatment with tofogliflozin markedly reduced hepatic fibrosis at both doses (10 and 20 mg/kg), as well as the number of α-smooth muscle actin^+^ myofibroblasts, which was increased in CCl_4_-treated rats (Figure 4A,B). These ameliorations in the fibrotic phenotypes coincided with reduced hepatic mRNA levels of profibrotic genes, including *Col1a1*, *Acta2*, *Pdgfrb*, and *Tgfbr1* (Figure 4C–F). Moreover, tofogliflozin treatment also reduced hepatic transforming growth factor-β1 (TGF-β1) production and attenuated the phosphorylation of SMAD2/3 in CCl_4_-treated rats (Figure 4G,H). In addition, the CCl_4_-treated rats showed an increase in hepatic mRNA expression of matrix metalloproteinases (MMPs) (*Mmp2*, *Mmp9,* and *Mmp13*) and tissue inhibitors of metalloproteinases (TIMPs) (*Timp1* and *Timp2*). Treatment with tofogliflozin reduced the expression of *Mmp13* and *Timp1* but did not affect *Mmp2*, *Mmp9*, and *Timp2* expression (Figure 4I,J and Appendix A). Western blotting analysis also showed reduced protein levels of MMP-13 and TIMP-1 following tofogliflozin treatment (Figure 4K).

### 3.6. Tofogliflozin Suppresses Ac-hHSC-Stimulated Capillarization and Vasoconstrictive Activity in hLSECs

We evaluated the effect of tofogliflozin on the acquisition of angiogenic and vasoconstrictive activities in hLSECs cocultured with Ac-hHSCs (Figure 5A). As shown in Figure 5B, *CD34* mRNA expression in hLSECs was significantly increased when cocultured with Ac-hHSCs versus when monocultured, with expression levels approaching those of hUVECs. This increase in CD34 expression was also found at the protein level, indicating that coculture with Ac-hHSCs induced capillarization in hLSECs (Figure 5C). Afterward, SGLT2 expression was compared among several types of human cell lines. *SGLT2* mRNA expression was slightly detected in primary human hepatocytes or Ac-hHSCs, but higher levels were seen in hLSECs (Figure 5D). In hLSECs, *SGLT2* mRNA expression was significantly upregulated under coculture conditions with hHSCs compared with a monoculture (Figure 5D). Treatment with tofogliflozin significantly inhibited the Ac-hHSC-stimulated upregulation of CD34 in a dose-dependent manner (Figure 5E). Expression of vascular cell adhesion molecule 1 (*VCAM1*), a modulator of sinusoidal capillarization, was reduced by treatment with tofogliflozin in hLSECs cocultured with Ac-hHSCs (Figure 5F) [25]. Conversely, coculture with Ac-hHSCs decreased the expression of *CD32b*, a physiological marker of LSEC; this was ameliorated by tofogliflozin treatment (Figure 5G).

Coculture with Ac-hHSCs increased the mRNA expression of vasoconstrictive markers, including *ET1*, cyclooxygenase 1 (*PTGS1*), and thromboxane A synthase 1 (*TBXAS1*), in hLSECs (Figure 5H and Appendix A). Notably, treatment with tofogliflozin significantly reduced the expression of *ET1* but not of *PTGS1* and *TBXAS1* (Figure 5H and Appendix A). In contrast, coculture with Ac-hHSCs decreased NO production and phosphorylation of eNOS in hLSECs; this was ameliorated by tofogliflozin treatment (Figure 5I,J). Moreover, tofogliflozin increased the expression of Caveolin-1, a marker of fenestration, which was decreased during coculture with Ac-hHSCs in hLSECs (Figure 5J).

In cirrhosis, reduced NO bioavailability may result from a decrease in NO production by LSECs and impaired removal of oxidative stress [26]. Thus, we measured the expression of classic antioxidant enzymes to assess the effect of tofogliflozin on oxidative stress in hLSECs. The mRNA expressions of glutathione peroxidase (GPX) 1 and 4, superoxide dismutase (SOD) 1, and catalase (CAT) were decreased in hLSECs cocultured with Ac-hHSCs. Tofogliflozin ameliorated these decreases (Figure 5K). In whole liver tissue of CCl_4_-treated rats, these antioxidant effects were insufficient, suggesting that tofogliflozin mainly targeted LSECs (Appendix A).

### 3.7. Effects of Tofogliflozin on hLSEC-Stimulated Contractive, Profibrogenic, and Proliferative Activities of Ac-hHSCs

Ac-HSCs are key mediators of PH and liver fibrosis in liver cirrhosis [3,4,5,7]. Thus, we evaluated the effect of tofogliflozin on the properties of Ac-hHSCs cocultured with hLSECs. We first explored the effect of tofogliflozin on the contraction of Ac-hHSCs using a collagen gel assay. As shown in Figure 6A,B, Ac-hHSCs showed clear contractile properties, which were enhanced in coculture with hLSECs. Notably, tofogliflozin reduced the contraction of Ac-hHSCs in coculture with hLSECs (Figure 6A,B). Moreover, tofogliflozin increased cGMP production in Ac-hHSC in a dose-dependent manner, which was decreased by coculture with hLSECs (Figure 6C). Meanwhile, tofogliflozin had no effect on contraction and cGMP production in monocultured Ac-hHSCs (Figure 6A–C). These findings indicate that tofogliflozin did not directly act on Ac-hHSC contractility yet exerted its action on hLSECs in a NO-dependent manner. In support of this, in Ac-HSC coculture with hLSECs, tofogliflozin increased the activity of soluble guanylyl cyclase (sGC), a primary sensor of NO to synthesize cGMP, but not in monoculture (Figure 6D). In contrast, tofogliflozin did not affect the activity of phosphodiesterase 5A1 (PDE5A1), an enzyme that breaks down cGMP, in both monoculture and coculture with hLSECs (Figure 6E).

We determined the downstream effect of myofibroblast transdifferentiation, specifically fibrosis molecule production. Similar to its effect on the contractile properties, tofogliflozin did not affect the mRNA expression of *COL1A1*, *ACTA2*, and *TGFB1* in the monocultured hHSCs (Appendix A). Conversely, tofogliflozin significantly reduced the expression of these profibrogenic markers, which were increased by coculture with hLSECs (Figure 6F). We also found that tofogliflozin suppressed SMAD2/3 phosphorylation (Figure 6G) and attenuated the proliferative activity of hHSC coculture with hLSECs, whereas it did not affect monocultured hHSCs (Figure 6H and Appendix A). Consistently, extracellular signal-regulated kinase1/2 (ERK1/2) phosphorylation was suppressed by treatment with tofogliflozin while attenuating cell proliferation (Figure 6I).

## 4. Discussion

Regardless of etiology, controlling PH and liver fibrosis in chronic liver disease is of paramount importance to prevent decompensation [1,2,3,4,5,11]. The present study provides strong evidence that tofogliflozin, an SGLT2-I, has potent beneficial effects on portal pressure, sinusoidal capillarization, and angiogenesis as well as liver fibrosis development in a CCl_4_-induced liver cirrhosis model. We elucidated the underlying mechanistic aspects of the crosstalk between LSEC and HSC.

Increasing evidence has demonstrated the antifibrotic effect of SGLT2-Is in both patients and animal models with MASLD [15,27,28,29]. However, there is limited direct evidence to assess the benefits of SGLT-2Is on PH and liver fibrosis in liver cirrhosis, although several clinical trials are ongoing to explore this potential application of SGLT2-Is [30,31]. Our results found no changes in serum glucose in nondiabetic rats with cirrhosis treated with tofogliflozin, indicating that the aforementioned effects of tofogliflozin are independent of its antidiabetic action. Moreover, although several reports have described a significant antihypertensive effect of SGLT2-Is, tofogliflozin treatment did not alter MAP and HR in the rats of our study [32,33]. This suggests that the reduction in portal pressure with tofogliflozin administration is a separate consequence of its effect on systemic circulation.

The mechanistic interplay of LSECs and HSCs is a major factor in the pathogenesis of PH. Several studies have suggested that Ac-HSCs impair the LSEC phenotype and induce sinusoidal capillarization and remodeling by secreting vasoactive substances [4,5,6,7,8,9,10]. In line with these findings, our cell-based analysis demonstrated that coculture with Ac-hHSCs induced hLSEC capillarization, evidenced by the increased CD34 and VCAM1 expression and decreased CD32b expression. Notably, we found that SGLT2 expression was higher in LSECs than in hepatocytes or Ac-hHSCs, and its expression increased as capillarization progressed. SGLT2 is mainly expressed in the renal proximal tubules, but many studies have recently revealed SGLT2 expression in vascular endothelial cells, as evidenced by the cardiovascular protective effects of SGLT2-Is [16,21,34]. Thus, it is possible that tofogliflozin predominantly affected capillaries in hLSECs through SGLT2 in the cirrhotic liver. However, the exact mechanism by which SGLT2 expression is increased during capillarization needs to be further elucidated.

We found that tofogliflozin attenuated capillarization and angiogenesis, as well as restored fenestration in hLSEC, accompanied by reduced ET-1 expression and increased eNOS-mediated NO production. ET-1 is a known potent endothelium-derived vasoconstrictor [35]. After a liver injury, Ac-HSCs are responsible for increased ET-1 synthesis by LSECs, which contributes to significant vasoconstriction and PH [4,5,6,7,8,9,10,36,37]. Indeed, plasma endothelin levels are increased in cirrhosis and correlate with the severity of liver disease and portal pressures [38,39]. Alternatively, the NO-cGMP pathway is a main regulator of sinusoidal tone via vasodilation and plays a key role in reducing portal pressure [3,4,5,6,7,8,9,10]. NO is generated from L-arginine by eNOS in LSECs and diffuses into neighboring HSCs, where it binds to and activates sGC and catalyzes the conversion of guanosine triphosphate to cGMP, eventually causing vascular dilation with a decrease in intracellular calcium ions [3,4,5,6,7,8,9,10]. Studies have demonstrated that LSEC dysfunction in the cirrhotic liver caused a decrease in NO release, leading to impaired vasodilation in the hepatic microcirculation and sinusoidal PH [3,4,5,6,7,8,9,10,40]. Our study focused on antioxidant capacity, specifically regarding the mechanism behind the effect of tofogliflozin on capillarized hLSECs. Numerous studies have reported that oxidative stress promotes endothelial dysfunction and decreases NO bioavailability. In a recent report, LSECs could be impaired by autophagy-deficiency-induced oxidative damage and promoted capillarization, increased ET-1 expression, decreased NO bioavailability, and liver fibrogenesis through HSC activation [41]. Meanwhile, SGLT2-Is have been reported to possess antioxidative properties by increasing the levels of antioxidants such as GPXs, SODs, and CAT, thus protecting the endothelium [42,43]. Accordingly, these markers were increased in hLSECs after treatment with tofogliflozin in our study. Notably, the increased expression of these antioxidant factors was not observed in the entire liver, showing that tofogliflozin acts mainly on capillarized hLSECs.

Since tofogliflozin suppressed the impairment of hLSEC, the functional properties of Ac-hHSCs were attenuated. Ac-HSCs are important promotors of liver cirrhosis-related PH through the contractile phenotype [4,5,7,10,37]. HSC contractility is mainly mediated by ET-1 via the Rho/ROCK pathway [37,44]. In our gel contraction assay, tofogliflozin inhibited Ac-hHSC contraction in a coculture with hLSECs but not in a monoculture. Thus, tofogliflozin also contributes to the reduction in portal pressure by reducing LSEC dysfunction, thereby indirectly acting on Ac-HSCs. Furthermore, we found that tofogliflozin induced an increase in cGMP production with sGC activity in Ac-hHSCs cocultured with hLSECs in parallel with increased NO production in hLSECs. Maintaining a normal LSEC phenotype also helps suppress the activation of HSC from quiescence via a NO-dependent sGC/cGMP pathway, which may be relevant in chronic liver diseases such as alcoholic liver injury [9]. Several related studies have demonstrated that NO can inhibit the proliferation and migration of Ac-HSCs, as well as induce HSC apoptosis [45,46]. In agreement with this functional evidence, we speculate that tofogliflozin inhibits profibrogenic and proliferative activities in capillarized hLSEC-stimulated hHSCs via the NO/sGC/cGMP pathway.

We found that treatment with tofogliflozin reduced hepatic enzymes and attenuated Kupffer cell infiltration in CCl_4_-induced cirrhotic rats. LSEC capillarization and dysfunction have also been implicated in Kupffer cell activation and hepatic inflammation [47]. In fact, eNOS-deficient mice exhibit an accelerated hepatic inflammatory response, while pharmacologically improving NO/cGMP signaling prevents hepatic inflammation in MASLD models [48]. Thus, we assume that tofogliflozin attenuates hepatic inflammation, at least in part, by suppressing LSEC capillarization and dysfunction. Other in vitro studies have suggested that some SGLT2-Is act directly on macrophages to promote a shift from inflammatory M1 macrophages toward M2-dominant macrophages, and thus, further investigation of the direct effect of tofogliflozin on Kupffer cells is needed [49].

Our study has some limitations. First, we found that tofogliflozin had a preventive effect on portal pressure elevation and the development of hepatic fibrosis alongside CCl_4_ exposure. In practice, however, pharmacologic treatment is usually given when PH has already developed. Thus, further investigation is needed using a model of drug administration at the stage of advanced cirrhosis. Second, it is necessary to prove whether similar effects can be reproduced with other types of SGLT2-Is. In particular, a recent study found that empagliflozin suppresses liver fibrosis by regulating different pathways (Galectin-1/Neuropilin-1) related to LSEC and HSC communication [50]. Additional analyses are required to evaluate whether this mechanism plays a role in the inhibitory effect of tofogliflozin on PH and hepatic fibrosis. Third, this study did not evaluate the effect of tofogliflozin on the NO/cGMP signaling in peripheral blood vessels, which is responsible for changes in peripheral vascular resistance. Recent findings suggest that changes in cGMP availability may better elucidate the contrasting findings of intrahepatic vasoconstriction and peripheral systemic vasodilation rather than simply focusing on NO availability [51]. In cirrhosis, there is evidence that hepatic cGMP decreases, leading to vasoconstriction, whereas in peripheral vessels, cGMP increases, leading to vasodilation [51]. In our result, CCl_4_-induced cirrhotic rats showed a decreased MAP. The results indicate peripheral vasodilation in these rats, suggesting a possible increase in cGMP and a decrease in PDE-5 in peripheral vessels. Meanwhile, MAP was not changed by treatment with tofogliflozin, presuming that tofogliflozin might have a limited effect on peripheral vascular resistance. To elucidate this effect and the detailed mechanism, it will be necessary to compare the expression of cGMP, sGC, and PDE-5 in peripheral vascular tissues in the present model as a subject for future investigation.

In conclusion, this was the first study to show that tofogliflozin, an SGLT2-I, delays PH, hepatic inflammation, and fibrosis progression independently of its antidiabetic effect in the CCl_4_-induced rat cirrhosis model. This effect of tofogliflozin is mediated by its antioxidant capacity, which suppresses LSEC capillarization and ET-1 expression, as well as activates the NO-dependent sGC/cGMP pathway in HSCs. Notably, tofogliflozin did not cause harmful hypotension. As a clinically available drug with limited toxicity, tofogliflozin may be an emerging therapeutic candidate for PH in liver cirrhosis.

## 5. Supplementary Materials and Methods

### 5.1. Biochemical Analysis

Serum AST and ALT were measured using the Rat AST ELISA kit (ab263883, Abcam, Cambridge, UK) and Rat ALT ELISA kit (ab285264, Abcam), respectively. Serum and urine glucose, BUN, and sCr levels were measured via UV absorption spectrophotometry.

### 5.2. Transforming Growth Factor-β1 (TGF-β1) Level

After equalizing the protein concentration of the frozen liver samples, hepatic TGF-β1 level was measured using the TGF-β1 ELISA kit (Proteintech, Rosemont, IL, USA), according to the manufacturer’s instructions.

### 5.3. Vascular Endothelial Growth Factor (VEGF)-A, Platelet-Derived Growth Factor (PDGF)-BB, and Von Willebrand Factor (VWF) Levels

Hepatic VEGF-A, PDGF-BB, and VWF levels were measured using the VEGF-A Rat ELISA Kit (Thermo Fisher Scientific), PDGF-BB ELISA KIT (Proteintech), and Rat von Willebrand Factor ELISA Kit (Novus Biologicals) according to the manufacturer’s instructions.

### 5.4. Histological and Immunohistochemical Analyses

Liver specimens were fixed in 10% formalin, embedded in paraffin, divided into 5-μm sections, and then stained with hematoxylin and eosin (H&E) and Sirius-Red (performed at Narabyouri Research Co., Nara, Japan). For immunohistochemical staining, liver tissue sections were blocked for 30 min following deparaffinization and antigen retrieval, then incubated overnight at 4 °C with mouse monoclonal CD68 antibody (1:100; GTX41868, GeneTex, Irvine, CA, USA). The sections were washed thrice with phosphate-buffered saline and subsequently incubated with a goat antimouse IgG (H+L) HRP-conjugated secondary antibody (1:2000; 62-6520, Thermo Fisher Scientific, Waltham, MA, USA) for 30 min at room temperature. The slides were developed with DAB until the signal clearly appeared, and the nuclei were stained with hematoxylin for 5 min.

For immunofluorescence, the liver sections were deparaffinized, rehydrated, and blocked in a manner similar to that used for immunohistochemical staining. Mouse monoclonal alpha smooth muscle actin (α-SMA) (ab7817, Abcam, Cambridge, UK) and rabbit monoclonal CD34 (ab81289, Abcam) were used as primary antibodies. Following overnight incubation at 4 °C, the immunofluorescence detection of the primary antibodies was performed using Alexa Fluorconjugated secondary antibodies (1:200; A-11020 and A-21206, Thermo Fisher Scientific) for 1 h at room temperature. The sections were mounted on Vectashield mounting medium with 4′,6-diamidino-2-phenylindole Fluoromount-G mounting medium for fluorescent nucleic acid staining (Vector Laboratories, Newark, CA, USA). Pathological analyses were independently performed by 2 pathologists in 10 random fields from each slide at a magnification of 400-fold. Semi-quantitative analysis was performed using ImageJ software version 64 (National Institutes of Health, Bethesda, MD, USA).

### 5.5. TMNK-1 Cocultured with LX-2

For coculture assays, TNMK-1 cells (3 × 105 cells) were seeded in the lower compartment, whereas LX-2 cells (1 × 105 cells) were cultured in the upper inserts of Transwell™ Multiple Well Plates (24 mm inserts, TC-treated, 0.4 µm pore size, 6-well cluster plate) (Corning, NY, USA). As the coculture medium, DMEM not containing FBS was used; for some assays, cells were incubated with different concentrations of tofogliflozin (0, 25, 50, and 100 μM).

### 5.6. The Real-Time Quantitative Polymerase Chain Reaction (RT-qPCR)

Total RNA was extracted from frozen liver tissue and whole cell lysates using an RNeasy mini kit (Qiagen, Tokyo, Japan) according to the manufacturer’s instructions. We performed cDNA synthesis using the High-Capacity RNA-to-cDNA™ Kit (Thermo Fisher Scientific) and RT-qPCR using the SYBR™-Green PCR Master Mix (Thermo Fisher Scientific) as per the manufacturer’s instructions. The fold change in gene expression was normalized with glyceraldehyde-3-phosphate dehydrogenase (GAPDH), and the relative fold change was calculated using the 2_−ΔΔCT_ method. All primer sequences used for RT-qPCR are listed in Appendix A.

### 5.7. Western Blotting Assays

Protein was extracted using RIPA lysis buffer (Sigma-Aldrich, St. Louis, MO, USA), plus Halt™ Protease and Phosphatase Inhibitor Cocktail (Thermo Fisher Scientific) from 200 mg of frozen liver tissue and 1 × 106 cultured cells. After normalizing the protein concentration to 50 μg, the proteins were separated using SDS-PAGE (Thermo Fisher Scientific) and then transferred to an Invitrolon™ PVDF membrane (Thermo Fisher Scientific). After sealing with 5% skimmed milk, the membranes were successively incubated with diluted primary antibodies and Amersham ECL IgG and HRP-linked F(ab)2 fragment (1:5000, Cytiva, Tokyo, Japan) as secondary antibodies. The primary antibodies included SMAD2/3 (#3102; 1:1000), phospho-SMAD2 (Ser465/Ser467)/3 (Ser423/425) (p-SMAD2/3) (#8828; 1:1000), eNOS (#32037; 1:1000), phospho-eNOS (Ser1177) (p-eNOS) (#9570; 1:1000), Caveolin-1 (#3238; 1:1000), extracellular signal-regulated kinase1/2 (ERK1/2) (#9102; 1:1000), phospho-ERK1/2 (Thr202/Tyr204) (#9101; 1:1000), GAPDH (#5174; 1:1000), and β-actin (Actin) (#4967; 1:10,000) from Cell Signaling Technology (Danvers, MA, USA); and Matrix Metalloproteinase-13 (MMP-13) (ab39012; 1:5000) from Abcam, CD34 (# MA1-10202; 1:1000), and Tissue Inhibitor of Metalloproteinase-1 (TIMP1) (#PA5-99559; 1: 1000) from Thermo Fisher Scientific. Finally, chemiluminescence was detected using a Clarity Western ECL Substrate (Bio-Rad, Hercules, CA, USA) with Bright™ CL1500 Imaging System (Thermo Fisher Scientific). Densitometric analysis was performed using ImageJ software version 64 (NIH).

### 5.8. ET-1, eNOS, NO, and Cyclic Guanosine Monophosphate (cGMP) Levels

Hepatic ET-1, eNOS, NO, and cGMP levels were measured using the Endothelin-1 ELISA kit (Enzo Life Sciences, Farmingdale, NY, USA), Rat eNOS ELISA Kit (Novus Biologicals, Centennial, CO, USA), cGMP ELISA kit (Cayman chemical, Ann Arbor, MI, USA), and QuantiChrom Nitric Oxide Assay Kit (BioAssay Systems, Hayward, CA, USA) according to the manufacturer’s instructions. Intracellular NO and cGMP levels in cultured LX-2 cells were measured using the same ELISA kit.

### 5.9. Collagen Gel Contraction Assay

A CytoSelect 24-Well Cell Contraction Assay Kit (Floating Matrix Model) (Cell Biolabs, SanDiego, CA, USA) was used in this study. According to the manufacturer’s instructions, a 0.5 mL cell contraction matrix containing 5 × 106 LX-2 cells was prepared in 24-well plates. Serum-free DMEM medium (1 mL) with different concentrations of tofogliflozin (0, 25, 50, and 100 μM) was added to the cell contraction matrix. For some assays, 1.5 × 107 of TMNK-1 cells were plated into 0.4 µm pore size transwell inserts. Gel contraction mediated by LX-2 was assessed by measuring the change in the gel area using ImageJ software (NIH) after 48 h.

### 5.10. Soluble Guanylyl Cyclase (sGC) and Phosphodiesterase 5A1 (PDE5A1) Activity Assay

LX-2 cells were monocultured or cocultured with TNMK-1 in medium containing tofogliflozin (0, 25, 50, and 100 μM) for 24 h. The sGC activity in the cultured LX-2 cells was determined as described previously.1 Briefly, cells were suspended in sGC lysis buffer (25 mM of Tris-HCl [pH 7.5], 250 of mM sucrose, 2 mM of EDTA, 5 mM of MgCl2, 100 μM of phenylmethylsulfonyl fluoride [PMSF], 10 μg/mL of leupeptin, and 10 μM of pepstain A). After centrifugation, the cytosolic fraction was mixed in sGC reaction buffer (25 mM of Tris-HCl [pH 7.5], 15 mM of MgCl2, 2 mM 3-isobutyl 1-methylxanthine, 15 mM of creatine phosphate, 6 units of creatine phosphokinase, and 2 mM of GTP) for 20 min at 30 °C, then stopped by heating at 100 °C for 3 min. To determine sGC activity, cGMP levels were measured using a cGMP ELISA kit (Cayman chemical) as described earlier. PDE5A1 activity was determined using a PDE5A Assay Kit (BPS Bioscience, SanDiego, CA, USA) according to the manufacturer’s instructions.

### 5.11. Cell Proliferation Assay

LX-2 cells were monocultured or cocultured with TNMK-1 in a medium containing tofogliflozin (0, 25, 50, and 100 μM) for 6, 12, 24, and 48 h. BrdU Cell Proliferation ELISA (Cosmo Bio, Tokyo, Japan) was used to evaluate cell proliferation according to the manufacturer’s protocol.

## Figures and Tables

**Figure 1 cells-13-00538-f001:**
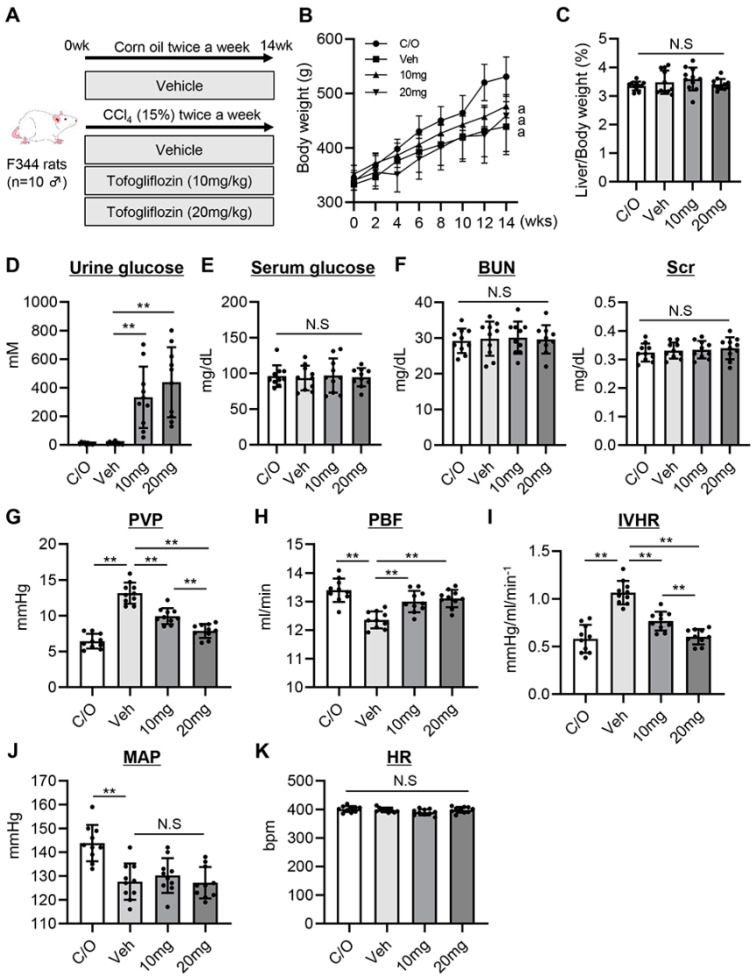
Effect of tofogliflozin on portal hemodynamics in cirrhotic rats. (**A**) In vivo experimental design. (**B**) Changes in the body weights during the experimental period. “a” indicates the significant difference (*p* > 0.05) between C/O and Veh, between C/O and 10 mg, between C/O and 20 mg at 14 wks. (**C**) Liver/body weight at the end of experiment. (**D**) Urine and (**E**) serum glucose levels. (**F**) Blood urea nitrogen (BUN) and serum creatinine (sCr) levels. (**G**) Portal vein pressure (PVP), (**H**) Portal blood flow (PBF), (**I**) Intrahepatic vascular resistance (IVHR; calculated as PVP/PBF), (**J**) Mean arterial pressure (MAP), and (**K**) Heart rate (HR) were measured at the end of experiment. Data are the mean ± SD (*n* = 10). ** *p* < 0.01, significant difference between groups determined by Student’s *t*-test. N.S, not significant; C/O, corn oil-injected negative control group; Veh, CCl_4_+ vehicle-treated group; 10 mg, CCl_4_+ tofogliflozin (10 mg/kg/day)-treated group; 20 mg, CCl_4_+ tofogliflozin (20 mg/kg/day)-treated group.

**Figure 2 cells-13-00538-f002:**
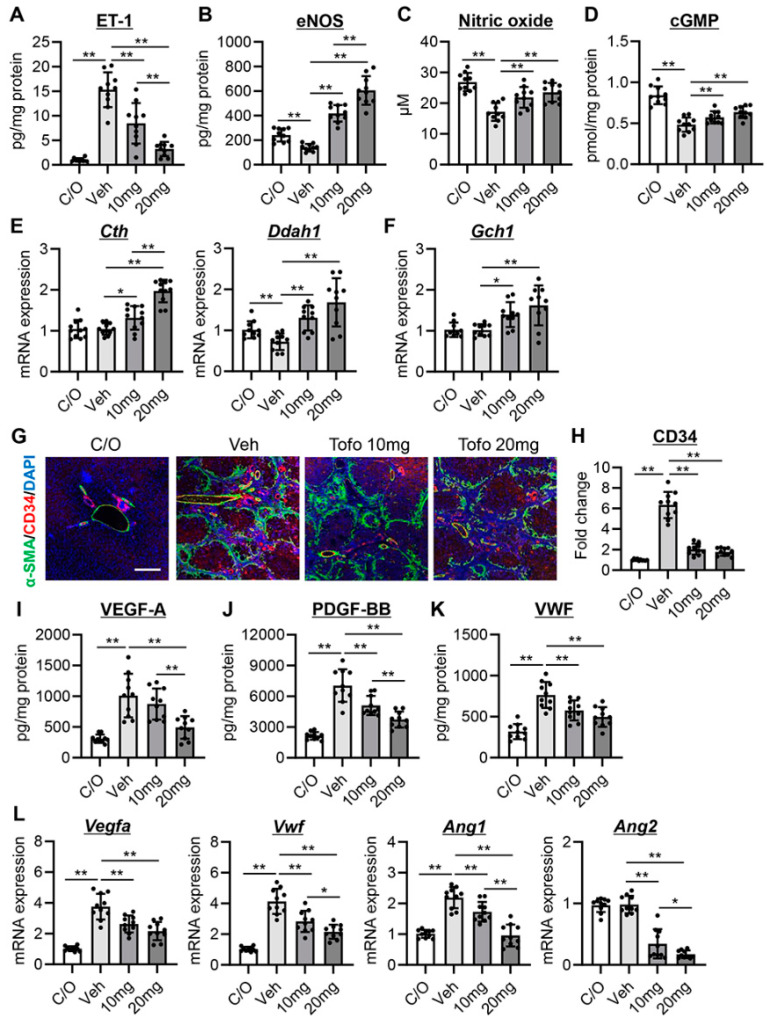
Effect of tofogliflozin on intrahepatic vasoconstriction, sinusoidal capillarization, and remodeling in cirrhotic rats. (**A**–**D**) Hepatic levels of (**A**) Endothelin-1 (ET-1), (**B**) Endothelial nitric oxide synthase (eNOS), (**C**) Nitric oxide, and (**D**) Cyclic guanosine monophosphate (cGMP). (**E**,**F**) Relative mRNA levels of vascular tone-related markers, including (**E**) *Cth* and *Ddah1* and (**F**) *Gch1* in the liver. (**G**) Double immunofluorescence with α-smooth muscle actin (α-SMA) and CD34 in the liver. Scale bar; 50 μm. (**H**) Quantification of CD34+-neovascularization. (**I**–**K**) Hepatic levels of (**I**) vascular endothelial growth factor (VEGF)-A, (**J**) Platelet-derived growth factor (PDGF)-BB, and (**K**) von Willebrand factor (VWF). (**L**) Relative mRNA levels of proangiogenic markers, including *Vegfa*, *Vwf*, Angiogenin (*Ang*)*1*, and *Ang2* in the liver. (**E**,**F**,**L**) *Gapdh* was used as an internal control for qRT-PCR. Quantitative values are indicated as fold changes to the values of C/O group. Data are the mean ± SD (*n* = 10). * *p* < 0.05, ** *p* < 0.01, significant difference between groups determined by Student’s *t*-test. N.S, not significant; C/O, corn oil-injected negative control group; Veh, CCl_4_+ vehicle-treated group; 10 mg, CCl_4_+ tofogliflozin (10 mg/kg/day)-treated group; 20 mg, CCl_4_+ tofogliflozin (20 mg/kg/day)-treated group.

**Figure 3 cells-13-00538-f003:**
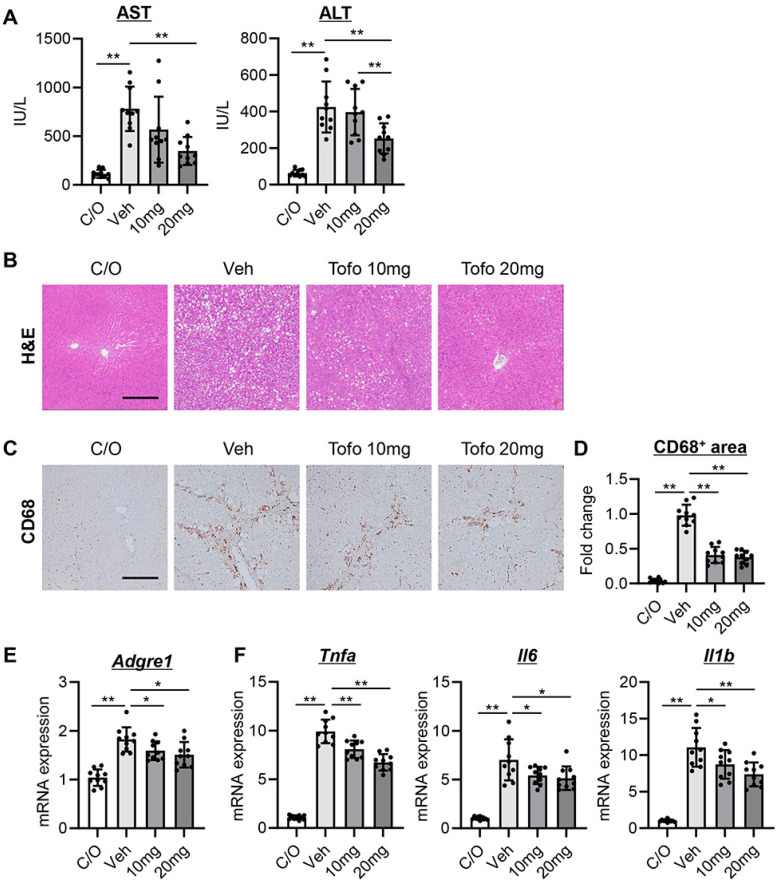
Effect of tofogliflozin on hepatic inflammation in cirrhotic rats. (**A**) Serum levels of aspartate transaminase (AST) and alanine transaminase (ALT). (**B**,**C**) (**B**) Hematoxylin and eosin (H&E) and (**C**) CD68 staining in the liver. Scale bar; 50 μm. (**D**) Quantification of CD68+-Kupffer cells. (**E**,**F**) Relative mRNA levels of (**E**) *Adgre1* and (**F**) *Tnfa*, *Il6,* and *Il1b* in the liver. *Gapdh* was used as an internal control for qRT-PCR. Quantitative values are indicated as fold changes to the values of (**D**) Veh group or (**E**,**F**) C/O group. Data are the mean ± SD (*n* = 10). * *p* < 0.05, ** *p* < 0.01, significant difference between groups determined by Student’s *t*-test. N.S, not significant; C/O, corn oil-injected negative control group; Veh, CCl_4_+ vehicle-treated group; 10 mg, CCl_4_+ tofogliflozin (10 mg/kg/day)-treated group; 20 mg, CCl_4_+ tofogliflozin (20 mg/kg/day)-treated group.

**Figure 4 cells-13-00538-f004:**
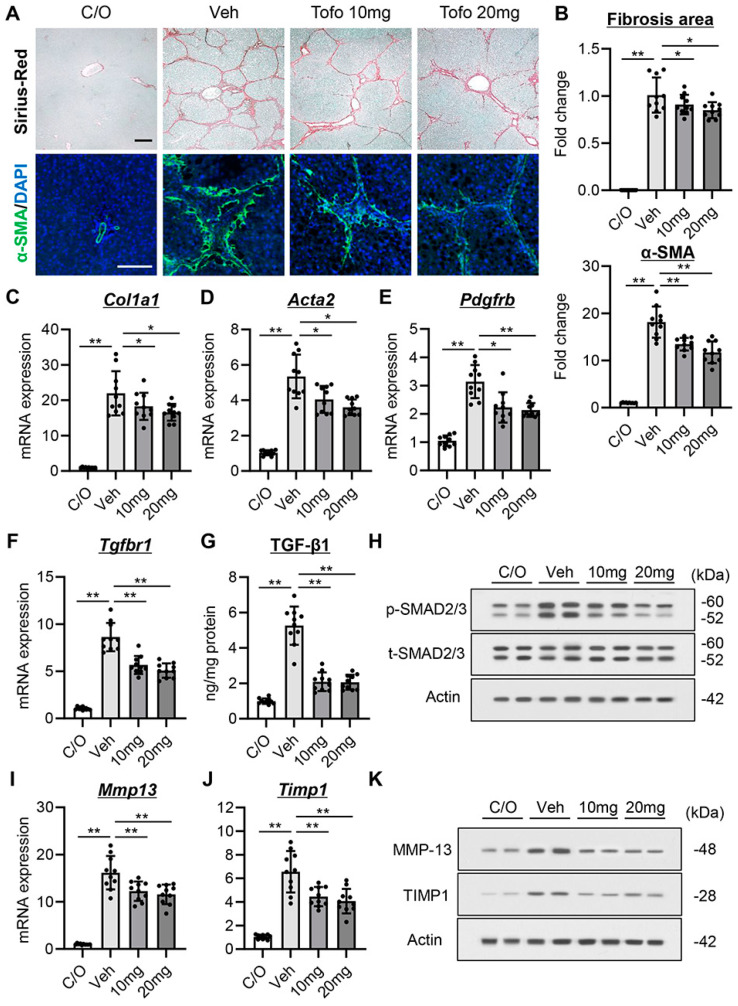
Effect of tofogliflozin on hepatic fibrosis in cirrhotic rats. (**A**) Sirius-Red staining and immunofluorescence with α-SMA. Scale bar; 50 μm. (**B**) Quantification of Sirius-Red-stained fibrotic area and α-SMA+-hepatic stellate cells. (**C**–**F**,**I**,**J**) Relative mRNA levels of (**C**) *Col1a1*, (**D**) *Acta2*, (**E**) *Pdgfrb*, (**F**) *Tgfbr1*, (**I**) *Mmp13,* and (**J**) *Timp1* in the liver. *Gapdh* was used as an internal control for qRT-PCR. (**G**) Hepatic level of transforming growth factor-β (TGF-β1). (**H**,**K**) Western blots for (**H**) the phosphorylation of SMAD2/3 and (**K**) The protein levels of matrix metalloproteinase (MMP)-13 and tissue inhibitor of metalloproteinase (TIMP) 1 in the liver. Actin was used as the loading control. Quantitative values are indicated as fold changes to the values of C/O group. Data are the mean ± SD (*n* = 10). * *p* < 0.05, ** *p* < 0.01, significant difference between groups determined by Student’s *t*-test. N.S, not significant; C/O, corn oil-injected negative control group; Veh, CCl_4_+ vehicle-treated group; 10 mg, CCl_4_+ tofogliflozin (10 mg/kg/day)-treated group; 20 mg, CCl_4_+ tofogliflozin (20 mg/kg/day)-treated group.

**Figure 5 cells-13-00538-f005:**
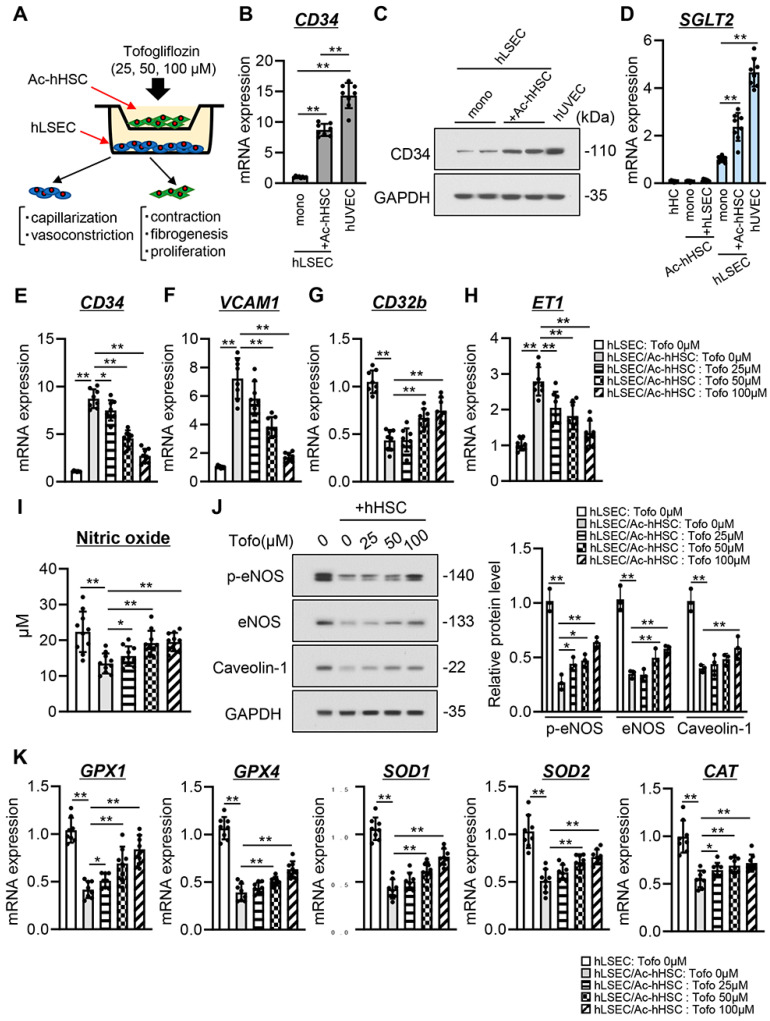
Effect of tofogliflozin on the dysfunction of LSEC co–cultured with activated HSC. (**A**) In vitro coculture system of human liver sinusoidal endothelial cell (hLSEC) and activated human HSC (Ac–hHSC). (**B**,**C**) (**B**) Relative mRNA level of *CD34* and (**C**) Western blots for CD34 protein expression in hLSEC under monoculture and coculture with Ac–hHSC. Expression level in human umbilical vascular endothelial cell (hUVEC) was used as positive control. (**D**) Relative mRNA level of sodium glucose transporter 2 (*SGLT2*) in human hepatocytes (hHC), Ac–hHSC, hLSEC, and hUVEC. Ac–hHSC and hLSEC were monocultured and cocultured with each other. (**E**–**H**) Effect of tofogliflozin (Tofo) on the expression of (**E**) *CD34*, (**F**) vascular cell adhesion molecule 1 (*VCAM1*), (**G**) *CD32b*, and (**H**) *ET1* in Ac–HSC-stimulated hLSEC. (**I**) Effect of tofogliflozin on intracellular nitric oxide production in Ac–HSC–stimulated hLSEC. (**J**) Western blots (left panel) and quantification (right panel) for the phosphorylation of eNOS and the protein expression of Caveolin–1 in Ac–HSC–stimulated hLSEC. (**K**) Effect of tofogliflozin on the anti–oxidant markers expression in Ac–HSC–stimulated hLSECs. *GAPDH* was used as internal control for qRT–PCR. GAPDH was used as the loading control for Western blotting. (**B**,**D**–**H**,**J**,**K**) Quantitative values are indicated as fold changes to the values of monocultured LSEC without tofogliflozin treatment (hLSEC: Tofo 0 μM) group. (**B**,**D**–**I**,**K**) Data are the mean ± SD (*n* = 8). * *p* < 0.05, ** *p* < 0.01, significant difference between groups by Student’s *t*–test. (**J**) Data are the mean ± SD (*n* = 3). * *p* < 0.05, ** *p* < 0.01, significant difference between groups determined by Mann–Whitney U test.

**Figure 6 cells-13-00538-f006:**
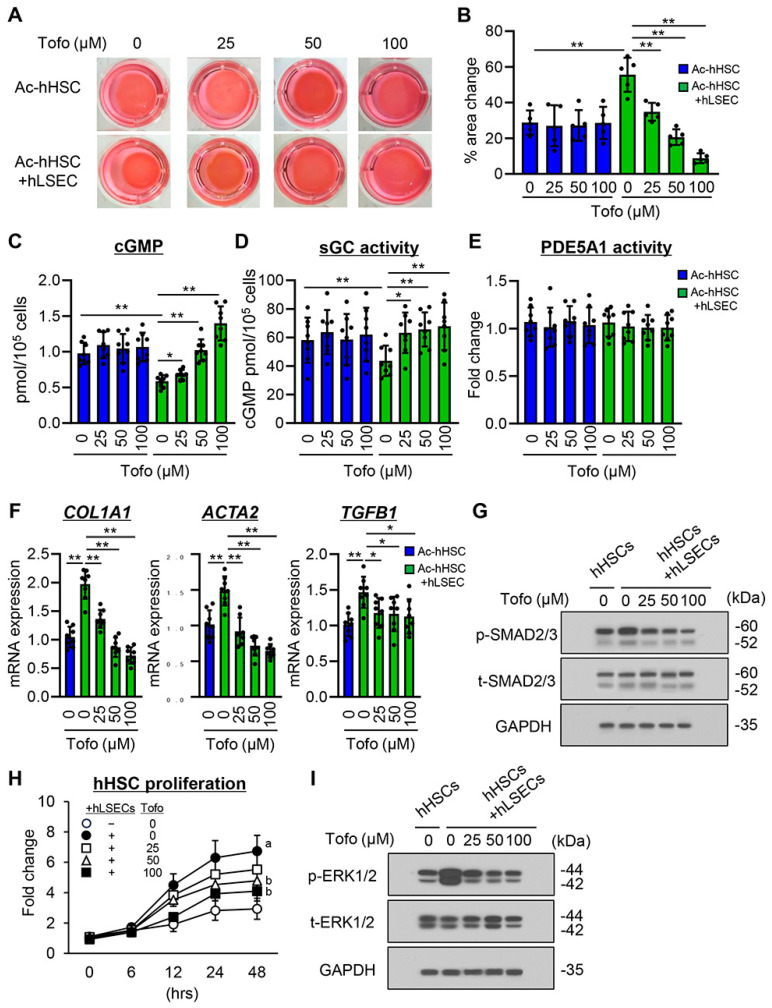
Effect of tofogliflozin on the phenotypes of activated HSC co–cultured with hLSEC. (**A**) Representative images of collagen gel mixed with activated human HSC (Ac–hHSC) under monoculture and co–culture with human liver sinusoidal endothelial cell (hLSEC). Both groups were pretreated with tofogliflozin (Tofo) (0, 25, 50, and 100 μM). (**B**) Statistical analysis of change in gel area. (**C**–**E**) Effect of Tofo on (**C**) intracellular cGMP production, (**D**) soluble guanylyl cyclase (sGC) activity, and (**E**) phosphodiesterase 5A1 (PDE5A1) activity in Ac–hHSC under monoculture and coculture with hLSEC. (**F**) Effect of Tofo on the expression of the profibrogenic markers in Ac–hHSC cocultured with hLSEC. (G and I) Western blot for the phosphorylation of (**G**) SMAD2/3 and (**I**) ERK1/2 in Ac–hHSC cocultured with hLSEC. (**H**) Effect of Tofo on cell proliferation in Ac–hHSC cocultured with hLSEC. *GAPDH* was used as internal control for qRT–PCR. GAPDH was used as the loading control for Western blotting. (**B**–**F**,**H**) Quantitative values are indicated as fold changes to the values of monocultured Ac–hHSC without tofogliflozin treatment group. Data are the mean ± SD ((**B**); *n* = 5, (**C**–**F**,**H**); *n* = 8). * *p* < 0.05, ** *p* < 0.01, significant difference between groups determined by Student’s *t*–test.

## Data Availability

The raw data supporting the conclusions of this article will be made available by the authors on request.

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
