# Peer review of "Tofogliflozin Delays Portal Hypertension and Hepatic Fibrosis by Inhibiting Sinusoidal Capillarization in Cirrhotic Rats"

_cells, 2024, doi:10.3390/cells13060538_

Round 1

Reviewer 1 Report

Comments and Suggestions for Authors

In this experimental study in a rat cirrhosis model, Tofogliflozin Is shown to reduce portal hypertension and hepatic fibrosis.

1.       Were the experiments repeated to verify the results

2.       Who read the pathology read-outs- single or multiple persons 

Comments on the Quality of English Language

None

Author Response

We thank the Reviewer for his/her positive evaluation of our work.

In this experimental study in a rat cirrhosis model, Tofogliflozin Is shown to reduce portal hypertension and hepatic fibrosis.

  1. Were the experiments repeated to verify the results

Answer

We appreciate the reviewer for kind comment. All experiments were performed a minimum of two times unless otherwise stated. We added this sentence in “Statistical analyzes” part in the revised manuscript (please see line 130).

  1. Who read the pathology read-outs- single or multiple persons

Answer

We apologize the reviewer for insufficient description. Pathological analyzes were independently performed by 2 pathologists in 10 random fields from each slide at a magnification of 400-fold. We added this information in the revised manuscript (please see line 528).

Reviewer 2 Report

Comments and Suggestions for Authors

The research work by Asada S., et al., is a preclinical work of exceptional originality, importance and scientific quality with potential clinical applications.

A. Major Comments:

1.     (Line 98): Authors should make a short description on the procedure and means used for  portal and systemic hemodynamic measurements.

2.    (Line 98): The sentence looks incomplete. Please mention what was measured and by what methods and means.

3.    (Figure 1): Authors should indicate the level of significance betwen the 10 and the 20mg dose of tofogliflozin in figures G, H and I.

4.    (Figure 2): Authors must indicate in the text and show in this Figure and the other Figures, if there are any significant differences between the two tofogliflozin doses.

B.  Minor Comments:

1.      (Line 31): The presence of esophageal varices themselves is not a sign of liver decompensation. Variceal varices appear in 50–60% of patients with compensated cirrhosis and up to 85% in patients with decompensated cirrhosis. (Gastrointest Endosc 2007; 65: 82).

2.      (Line 136): Please write "PVP" instead of "PP".

Author Response

We thank the Reviewer for his/her positive evaluation of our work.

The research work by Asada S., et al., is a preclinical work of exceptional originality, importance and scientific quality with potential clinical applications.

  1. Major Comments:

  1. (Line 98): Authors should make a short description on the procedure and means used for portal and systemic hemodynamic measurements.

Answer

We apologize the reviewer for confusing description. In the original manuscript, we described the method of “in vivo hemodynamic evaluation” in Supplementary Materials and Methods. As the reviewer mentioned, however, this method should be described following “in vivo experimental protocol” in Materials and Methods section. We moved this part to the appropriate place. (please see line 102-113)

  1. (Line 98): The sentence looks incomplete. Please mention what was measured and by what methods and means.

Answer

We apologize the reviewer for insufficient description. We corrected this sentence as below (please see line 99).

then blood was collected from the aorta to measure serum levels of hepatic enzymes, kidney function test and glucose.

  1. (Figure 1): Authors should indicate the level of significance between the 10 and the 20mg dose of tofogliflozin in figures G, H and I.
  2. (Figure 2): Authors must indicate in the text and show in this Figure and the other Figures, if there are any significant differences between the two tofogliflozin doses.

Answer

We really appreciate the reviewer for providing kind suggestion. We added the significance difference between the two tofogliflozin doses in Figures 1G, 1H, 2A, 2B, 2E, 2I, 2J, 2L, and 3A, and for data where significant differences were found, the data referred to in the text.

  1. Minor Comments:

  1. (Line 31): The presence of esophageal varices themselves is not a sign of liver decompensation. Variceal varices appear in 50–60% of patients with compensated cirrhosis and up to 85% in patients with decompensated cirrhosis. (Gastrointest Endosc 2007; 65: 82).

Answer

We really thank the reviewer for pointing out the important issue and agree with the reviewer’s opinion. We changed to “esophagogastric variceal bleeding”.

  1. (Line 136): Please write "PVP" instead of "PP".

Answer

Thank you for the reviewer’s comment. We corrected these parts.

Reviewer 3 Report

Comments and Suggestions for Authors

This is a very well designed study. It showns that a SGLT-2-inhibitor has an unextepted effect on portal pressure, sinusoidal capillarization, and fibrogenesis in CCl4-induced liver damage.

The experiments are well described, and the conclusions are sound. My only point of criticism is that the role of the NO-cGMP is not adequately discussed. There is evidence that in liver cirrhosis intrahepatic cGMP is reduced leading to vasoconstriction, whereas in peripheral blood vessels cGMP sís increased leading to vasodilation. Consider the role of sGC and PDE-5. Did the authorts measure these parameters? If not, they should mention this hypothesis. For references, seen IJMS in the last two years. Endothelin may play a minor role.

Author Response

We thank the Reviewer for his/her positive evaluation of our work.

This is a very well designed study. It showns that a SGLT-2-inhibitor has an unextepted effect on portal pressure, sinusoidal capillarization, and fibrogenesis in CCl4-induced liver damage. The experiments are well described, and the conclusions are sound. My only point of criticism is that the role of the NO-cGMP is not adequately discussed. There is evidence that in liver cirrhosis intrahepatic cGMP is reduced leading to vasoconstriction, whereas in peripheral blood vessels cGMP is increased leading to vasodilation. Consider the role of sGC and PDE-5. Did the authors measure these parameters? If not, they should mention this hypothesis. For references, seen IJMS in the last two years. Endothelin may play a minor role.

Answer

We really appreciate the reviewer for generous suggestion. Recent findings suggest that changes in cGMP availability may better elucidate the contrasting findings of intrahepatic vasoconstriction and peripheral systemic vasodilation, rather than simply focusing on NO availability. Thus, in the present model, it would be extremely important to evaluate the NO/cGMP pathway in peripheral vessels, which is responsible for changes in peripheral vascular resistance. In our result, CCl4-induced cirrhotic rats expectedly showed a decreased mean arterial pressure (MAP) (Figure 1J). The results show peripheral vasodilation in these rats, suggesting a possible increase in cGMP and decrease in PDE-5 in peripheral vessels. Meanwhile, MAP in cirrhotic rats were not changed by treatment with tofogliflozin, suggesting that tofogliflozin might have a limited effect on peripheral vascular resistance (Figure 1J). Unfortunately, we did not measure the levels of cGMP, PDE-5 and sGC in peripheral blood vessels of experimental mice in this study. Although it is hypothesized that there are no significant changes with tofogliflozin administration, analysis of how these parameters are altered by tofogliflozin is an important topic for future study. We discussed this point with proper reference (Int J Mol Sci. 2021 Sep 26;22(19):10372.) in the revised manuscript (please see line 469-482).